# Extraction, Structure, and Pharmacological Activities of *Astragalus* Polysaccharides

**Jia Wang** [1,†] ⓘ, **Junying Jia** [2,†], **Li Song** [1], **Xue Gong** [1], **Jianping Xu** [1], **Min Yang** [1] and **Minhui Li** [1,3,*] ⓘ

1. Inner Mongolia Key Laboratory of Characteristic Geoherbs Resources Protection and Utilization, Baotou Medical College, Baotou 014060, Inner Mongolia, China; wwwmokcom@163.com (J.W.); hhhtsongli@126.com (L.S.); gongxue_2017@yeah.net (X.G.); xu_jianping1992@163.com (J.X.); yangmin_0406@aliyun.com (M.Y.)
2. Agricultural College, Inner Mongolia University for Nationalities, Tongliao 028000, Inner Mongolia, China; jjy509628@163.com
3. Pharmaceutical Laboratory, Inner Mongolia Institute of Traditional Chinese Medicine, Hohhot 010020, Inner Mongolia, China
* Correspondence: prof_liminhui@yeah.net; Tel.: +86-4727-1677-95
† These authors contributed equally to this work.

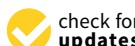

**Featured Application: This work will serve as a valuable resource for the research on APS.**

**Abstract:** The *Astragalus* polysaccharides (APS) are important bioactive components of Astragali Radix, the dry root of *Astragalus membranaceus*, which has been used in traditional Chinese medicine. In this review, the extraction conditions and extraction rates of APS are first compared for water, microwave-assisted, ultrasonic wave, and enzymatic hydrolysis extraction methods. Some studies have also shown that different methods can be combined to improve the extraction rate of APS. Subsequently, the chemical composition and structure of APS are discussed, as related to the extraction and purification method. Most studies have shown that APS is mainly composed of glucose, in addition to rhamnose, galactose, arabinose, xylose, mannose, glucuronic acid, and galacturonic acid. We also reviewed studies on the modification of APS using chemical methods, including sulfated modification using the chlorosulfonic acid–pyridine method, which is commonly used for chemical modification of APS. Finally, the pharmacological activities and mechanisms of action of APS are summarized, with a special focus on its immunoregulatory, antitumor, anti-inflammatory, and antiviral effects. This review will serve as a valuable resource for the research on APS.

**Keywords:** *Astragalus* polysaccharides; extraction; chemical composition; structural modification; pharmacological activity

## 1. Introduction

Astragali Radix (Huangqi) is the dry root of *Astragalus membranaceus* (Fisch.) Bge. and *A. membranaceus* (Fisch.) Bge. var. *mongholicus* (Bge.) Hsiao. It is a traditional Chinese medicine, which is described in the 2015 edition of Chinese Pharmacopoeia [1]. The use of Astragali Radix dates back to more than 2000 years ago, and it was recorded in *Shen Nong's Materia Medica*, written during the Han dynasty. Astragali Radix growing in the Inner Mongolia area is considered a genuine medicinal material and is famous for its high quality [2]. Since there is a substantial demand for Astragali Radix and its wild resources have been nearly exhausted, Astragali Radix for commercial purposes is currently mainly obtained by artificial cultivation. Astragali Radix produced in Inner

Mongolia ranks first in China and is mainly produced in Chifeng, Baotou, Ulanchabu, Bayannaoer, Alxa League, and Hohhot [3,4].

Studies conducted in the last two decades have provided significant insights into the pharmacological activities of different components of Astragali Radix, especially its polysaccharide fraction [5]. The *Astragalus* polysaccharides (APS) are important bioactive components of Astragali Radix, which has important pharmacological activities both in vivo and in vitro, such as immunomodulation, antitumor, anti-inflammatory, antiviral, antioxidant, anti-aging, and cardiovascular protection activities [6–8]. Its chemical composition and structure play an important role in the pharmacological activities of APS. Numerous studies have found that the structural modification of polysaccharides greatly enhances their biological activities. To further develop and utilize APS, studies have focused on the extraction process of APS since the 1980s. Currently, the commonly used methods include water, microwave-assisted, ultrasonic wave, and enzymatic hydrolysis extraction. In addition, several studies have suggested that a combination of these methods can increase the extraction rate of APS. At present, studies on the modification of APS mainly use chemical methods, including sulfation, phosphorylation, selenation, carboxymethylation, acetylation, alkylation, etc. In this paper, extraction, purification, the chemical composition, structural modification, and pharmacological functions of APS are reviewed to provide the basis for the development and clinical application of APS.

## 2. Extraction of *Astragalus* Polysaccharides

Polysaccharides are polar macromolecules, and different kinds of polysaccharides are obtained with different extraction methods. Most of the methods involve extraction with water or diluted alkali solutions as solvents. In recent years, the extraction of APS has been studied extensively, and some of the commonly used extraction methods are described below.

### 2.1. Water Extraction Methods

These methods are simple and easy to operate. Moreover, they are the most traditional extraction method for APS. The main factors affecting the extraction process are ranked in the following decreasing order of their effects: extraction temperature > material/liquid ratio > extraction time [9]. Zhu et al. [10] concentrated a crude water extract of APS to a certain volume using a rotary evaporator, then added absolute ethanol, allowed the mixture to stay overnight, and finally precipitated and centrifuged the extract. The results showed that the best solid-to-liquid ratio was 1:10, and the best extraction temperature and time were 80 °C and 2 h, respectively; under these conditions, the extraction rate reached 9.77%. Aqueous solutions of CaO and $Na_2CO_3$ have also been used to prepare APS crude extracts, and the results showed that the extraction yield was the highest (11.7%) with a CaO aqueous solution, which was 3.25 and 2.05 times higher than that obtained with water (3.6%) and a $Na_2CO_3$ aqueous solution (5.7%), respectively [11]. However, the temperature of boiling water is usually 100 °C, leading to two major disadvantages. The first one is the poor selectivity of the extraction method so that components such as flavonoids and saponins cannot subsequently be separated from APS. The second disadvantage is the waste of energy and resources, leading to low economic returns.

### 2.2. Microwave-Assisted Extraction Methods

Microwaves are characterized by strong penetration, high selectivity, and a high heating efficiency. The thermal effect of microwaves can break the cell wall and inactivate enzymes in the cell membrane; therefore, polysaccharides can be easily extracted from cells, and the yield can be effectively improved. Thus, microwaves can be employed in the extraction of APS. The following have been reported as the optimal extraction conditions for microwave-assisted extraction: the water/material ratio, 12:1; pH, regulated by saturated limewater, 9; and two doses of microwave radiation (300 W) for 10 min each. Under these conditions, the yield of the crude APS product was 14.6%, and the purity was 88.1% [12]. In another study using a microwave-assisted extraction method, the extraction rate of APS was 4.50%, with the APS content of 31.25%, indicating that this method was time and energy saving

and highly efficient [13]. Dong et al. [14] have reported the optimum enzymatic-microwave extraction conditions as follows: the liquid/solid ratio, 10:1; enzyme ratio, 57.6 U/g; and cellulase reaction time, 60 min, followed by 8 min of microwave irradiation (480 W). Under these conditions, the maximum extraction rate reached 16.07%, and the purity was up to 88.40%, which were considerably higher than those achieved by other extraction methods. Although microwave-assisted extraction methods can improve the extraction rate of polysaccharides, the effects of microwaves on the chemical structure and pharmacological activities of polysaccharides are still unclear and require further studies.

## 2.3. Ultrasonic Wave Extraction Methods

The cavitation effect of ultrasound leads to the rupture of the plant cell wall, thereby increasing the yield of polysaccharides. Ultrasonic extraction methods have been used to extract APS since they can shorten the extraction time and improve the extraction efficiency. A study on ultrasonic-assisted extraction of APS has reported the following as the optimal extraction conditions: the extraction time, 90 min; ultrasonic power, 250 W; and extraction temperature, 80 °C [15]. The effects of these factors on ultrasonic extraction were shown to decrease in the following order: ultrasonic power > extraction temperature > extraction time. The extraction of APS using ultrasonic waves in combination with microwaves resulted in an extraction rate as high as 4.25% under the following optimal extraction conditions: the extraction time, 150 s; microwave power, 120 W; and solid/liquid ratio, 1:25 [16]. When ultrasonic extraction was combined with cellulase hydrolysis to extract APS, the effects of the factors influencing the extraction rate were shown to decrease in the following order: ultrasonic time > ultrasonic temperature > enzyme amount > material/liquid ratio. The extraction rate was 24.12% under the following optimal conditions: the ultrasound time, 30 min; ultrasound temperature, 40 °C; solid/liquid ratio, 1:20; and enzyme amount, 10 mg [17]. Overall, an ultrasonic wave extraction method can be an important supplement to other methods.

## 2.4. Enzymatic Hydrolysis Extraction Methods

Cellulase can break down the plant cell wall, thereby releasing polysaccharides from cells without destroying the structure of polysaccharides. For APS extraction with cellulase, the optimum enzymatic hydrolysis time was 120 min, the ratio of the enzyme was 0.8%, and the hydrolysis temperature was 75 °C. Under these conditions, the extraction rate of APS was 9.78%, and that of total sugar was 50.2% [18]. After pretreatment with three different cellulase concentrations (0.3%, 0.4%, and 0.5%), the yield of APS obtained by a water extraction method increased by 314.8%, 392.6%, and 342.6%, respectively, compared with that obtained by the water extraction method alone [19]. When enzymatic hydrolysis was combined with a microwave method, the maximum extraction rate of APS reached 16.07%, and the purity was 88.40% under the following optimum extraction conditions: the liquid/solid ratio, 10:1; enzyme ratio, 57.6 U/g; and cellulase reaction time, 60 min, followed by 8 min of microwave irradiation (480 W) [14]. In conclusion, enzymatic hydrolysis of plant material can be employed as a pretreatment to improve the yield of APS.

## 3. Purification of *Astragalus* Polysaccharides

The purity of the extracts obtained with the above extraction methods is not sufficient for APS to be used for chemical composition and structure analyses. The extracted APS usually contains oligosaccharides, pigments, proteins, flavonoids, and other impurities. Therefore, purification must be carried out. The common purification methods employed are as follows: enzyme-Sevag, diethylaminoethyl-Sephadex A-25, and Sephadex G-100 [10]; a polyamide column and an AB-8 macroporous resin column [9,20]; X-5 macroporous resin [21]; chitosan flocculation [22]; and a type II ZTC1+1 natural clarifier [23]. The purified APS obtained by these methods can be used for subsequent chemical composition and structural analyses.

## 4. Chemical Composition and Structure of *Astragalus* Polysaccharides

The monosaccharide compositions of APS obtained from different plant varieties and habitats and using different methods are different. High-performance liquid chromatography (HPLC), gas chromatography (GC), mass spectrometry (MS), and nuclear magnetic resonance (NMR) are commonly used to analyze the chemical composition and structure of APS. Most studies have shown that APS is mainly composed of glucose, in addition to rhamnose, galactose, arabinose, and other monosaccharides (Figure 1). The main chain contains linked $\alpha$-(1→4) glucose residues. The relative molecular mass of APS is $5.6 \times 10^3$–$10^6$ Da (Table 1). Monosaccharide composition analysis of an APS sample by HPLC revealed that it was composed of rhamnose, glucose, galacturonic acid, and arabinose in a molar ratio of 1.19:72.01:5.85:20.95 [24]. Hydrolysis of another APS sample with trifluoroacetic acid to monosaccharides and analysis of the monosaccharide composition by HPLC revealed that this polysaccharide was composed of mannose, rhamnose, galacturonic acid, glucose, and arabinose in a molar ratio of 0.02:0.05:0.17:1.00:0.18 [25]. Yao et al. [26] determined the monosaccharide composition of APS by capillary GC, and the results showed that it was composed of arabinose, fructose, glucose, and mannose in a molar ratio of 1.00:3.24:7.00:0.46.

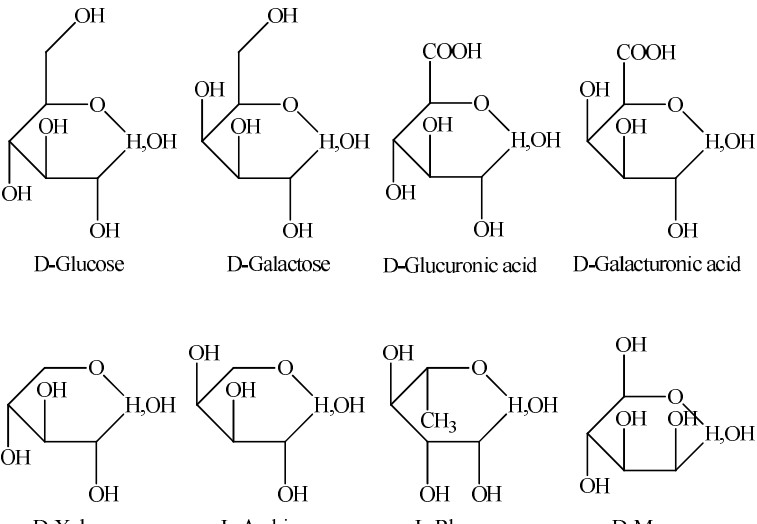

**Figure 1.** The monosaccharide structure of *Astragalus* polysaccharides.

The structures of APS isolated from different varieties of Astragali Radix are also different. Moreover, the chemical structure of polysaccharides influences their biological activities. Thus, Chen et al. [27] analyzed monosaccharides in APS by GC-MS, and the results revealed that the polysaccharide was mainly composed of L-rhamnose, L-arabinose, D-xylose, L-xylose, D-ribose, L-ribose, D-galactose, D-glucose, and D-mannose. Furthermore, monosaccharides extracted from a different *A. membranaceus* variety were different. The structure of APS was determined by GC-MS and infrared spectroscopy (IR), and the results showed that the main chain was mainly composed of glucose, xylose, and galactose, and the side chains were composed of glucose, arabinose, and galactose [28]. Furthermore, the branching point was composed of glucose, galactose, and arabinose, and the terminal residue was glucose. The APS powder for injection, developed by Panhua Pharmaceutical Co., Ltd., is mainly used as a chemotherapy or adjuvant therapy after radiotherapy in cancer patients. It has been reported that the main components of APS are alpha-1,4(1,6)-glucan, arabinose–galactose, rhamnose–galacturonic acid polysaccharide, and arabinose–galactoprotein polysaccharide [29]. These studies have laid a foundation for further analysis of the mechanisms of pharmacological activities of APS.

**Table 1.** The chemical composition of *Astragalus* Polysaccharides.

| Name | Molecular Mass (Da) | Monosaccharide Composition | Molar Ratio | Method | Reference |
|---|---|---|---|---|---|
| APS | $3.0 \times 10^5$ | L-Rhamnose, D-Xylose, D-Glucose, D-Galactose | 1:4:5:1.5 | GC; H-1, C-13 NMR | Fu et al. [30] |
| LMw-APS | $5.6 \times 10^3$ | Glucose, Galactose, Arabinose, Galactoside acid, Xylose | 10:1.3:1.7:0.95:1 | HPLC; GC | Qu [28] |
| APS | $2.1 \times 10^4$ | 1,4 Glucose backbone, 1,6 Glucose branched | | NMR | Niu et al. [31] |
| Rap-APS | $1.3 \times 10^6$ | Rhamnose, Arabinose, Glucose, galactose, Galactoside acid | 0.03:1.00:0.27:0.36:0.30 | GC-MS; 1H, 13C NMR | Yin et al. [32] |
| APS | $7.6 \times 10^6$ | L-Arabinose, D-Galactose, D-Galacturonic acid, D-Glucuronic acid | 18:18:1:1 | Electrophoresis; GC | Shimizu et al. [33] |
| APS | $3.6 \times 10^5$ | α-D-(1→4) glucose | | GC-MS; $^{13}C$NMR | Wang et al. [34] |
| | $1.1 \times 10^4$ | Rhamnose, Glucose, Galactose, Arabinoser | 1.19:72.01:5.85:20.95 | HPLC; IR; $^1$HNME | Li et al. [35] |
| MAPS-5 | $1.3 \times 10^4$ | α-D-(1→4) glucose | | GC-MS; IR; NMR | Lin et al. [36] |
| APS I | $4.8 \times 10^6$ | Arabinose, Xylose, Glucose | 0.54:1.00:18.14 | HPLC; GC; IR; NMR | Liu [37] |
| APS II | $8.7 \times 10^3$ | Arabinose, Xylose, Glucose | 0.23:1.00:29.39 | HPLC; GC; IR; NMR | Liu [37] |
| APS | $3.8 \times 10^4$ | Glucose, Galactose, Arabinose | 1.00:0.95:0.70 | HPLC | Liu et al. [38] |
| APS | $3.6 \times 10^4$ | α-(1→4)-D-glucose | | FTIR; AMLC; GLC- MS; NMR | Li et al. [39] |
| HAPS | | Rhamnose, Arabinose, Xylose, Mannose, Galactose, Glucose | 1.00:2.26:0.21:0.74:2.49:19.47 | HPLC; GC | Shan et al. [40] |
| APS | | Rhamnose, Arabinose, Xylose, Mannose, Galactose, Glucose | 1.00:4.34:3.92:1.95:11.41:20.52 | HPLC; GC | Shan et al. [40] |
| HAPS | $1.7 \times 10^6$ | Mannose, Glucose, Xylose, Arabinose, Glucuronic acid, Rhamnose | 0.06:28.34:0.58:0.24:0.33:0.21 | UPLC/ESI-Q-TOF-MS; FT-IR and NMR | Liao et al. [41] |
| APS | $2.0 \times 10^6$ | Mannose, Glucose, Xylose, Arabinose, Glucuronic acid, Rhamnose | 0.27:12.83:1.63:0.71:1.04:0.56 | UPLC/ESI-Q-TOF-MS; FT-IR and NMR | Liao et al. [41] |
| APP-2A | $2.3 \times 10^6$ | Rhamnose, Galactose, Arabinose, Glucose | 1.00:2.13:3.22:6.18 | FT-IR; GC; NMR | Pu et al. [42] |

Note: APS: *Astragalus* polysaccharides; GC: Gas chromatography; NMR: Nuclear magnetic resonance; HPLC: High-performance liquid chromatography; MS: Mass spectrometry; IR: Infrared spectroscopy.

### 5. Structural Modification of *Astragalus* Polysaccharides

Pharmacological activities of polysaccharides are closely related to their structures. Several studies have shown that structural modification of polysaccharides can change their pharmacological activities. Structural modification is usually performed using chemical, physical, enzymatic, and other effective methods for the improvement of nutraceutical and therapeutic functions of polysaccharides. Currently, studies on the modification of APS mainly employ chemical methods, including sulfation, phosphorylation, selenation, and carboxymethylation.

Sulfation of APS is a commonly used chemical modification method. It involves the reaction of polysaccharides, dissolved in a specific solvent, with corresponding sulfating reagents under certain conditions, leading to the introduction of sulfate groups into hydroxyl groups of side chains of polysaccharides (Figure 2). Sulfated polysaccharides can be obtained using the chlorosulfonic acid–pyridine (CSA–Pyr), sulfuric acid, and sulfur trioxide–dimethylacetamide or pyridine methods [43–45]. The CSA–Pyr method is the most popular, owing to its convenience, high yield, and high degree of sulfation. The sulfated APS obtained by the CSA–Pyr method had a better anti-inflammatory activity than did unmodified APS, in vitro and in vivo [46,47]. Sulfation of APS by the CSA–Pyr method also enhanced its antiviral effect [48]. Huang et al. [49] have found that compared with non-sulfated APS, sulfated APS could significantly increase the antibody titer and promote lymphocyte proliferation. Therefore, sulfated APS can be a candidate for a new immune adjuvant [50,51].

**Figure 2.** The reaction of APS sulfate by the chlorosulfonic acid–pyridine (CSA–Pyr) method.

Phosphorylation is a covalent modification of hydroxyl groups in side chains of polysaccharides with phosphate groups. Studies have shown that phosphorylation of polysaccharides can enhance their bioactivities. Phosphorylation of APS using the sodium tripolyphosphate–sodium trimetaphosphate method enhanced its antiviral effect against duck viral hepatitis [52]. Reaction with polyphosphoric acid under alkaline conditions resulted in a good antiviral activity of APS against porcine reproductive and respiratory syndrome virus [53].

Natural Se-containing polysaccharides have been found in several animals, plants, and microorganisms. As organic Se compounds, polysaccharides modified with Se can exhibit the physiological activities of both Se and polysaccharides. Moreover, the bioavailability of Se and its physiological functions as an essential trace element are effectively improved, while its toxicity and side effects are considerably lower than those of inorganic Se. Gong et al. [54] have reacted APS with a $SeOCl_2$ reagent and obtained a Se-containing APS, with a Se content of 16.820 mg/g. It has been reported that the inhibitory rate of tumor growth was 51.14% in the Se–APS group compared with 23.66% in the water control group, suggesting that combining APS with Se might enhance not only the tumor inhibitory effects of APS but also the antioxidant effect of Se [55,56]. It has been reported that a high Se content in Se-modified APS increases the antioxidant effect of APS [57].

Carboxymethyl groups are introduced into polysaccharide chains for complete carboxymethylation of polysaccharides. Carboxymethylation increases the negative charge of polysaccharide chains and their solubility in water. In addition, carboxymethylation has a strong effect on the bioactivity of polysaccharides. Yang et al. [58] prepared carboxymethyl-modified APS in the reaction with NaOH and $C_2H_3ClO_2$. The optimum reaction conditions were as follows: the reaction temperature, 65 °C; NaOH/$C_2H_3ClO_2$ ratio, 16:1; and reaction time, 3.5 h. The results showed

that the carboxymethyl-modified APS had the highest growth-promotion for microwave and immunological activities.

## 6. Pharmacological Activities of *Astragalus* Polysaccharides

Astragali Radix enhances immunity, as well as antioxidation, antiradiation, antitumor, antibacterial, and antiviral effects, and protects the cardiovascular and cerebrovascular systems, as well as the liver, kidneys, and lungs [59]. The *Astragalus* polysaccharides are important chemical components of Astragali Radix. Recent studies have indicated that APS has immunoregulatory, antitumor, anti-inflammatory, antiviral, antioxidant, anti-aging, and other biological activities.

### 6.1. Immunoregulatory Effects

The *Astragalus* polysaccharides not only enhance the function of immune organs and cells but also stimulate the release of cytokines, affect the nervous–endocrine–immune system network, and promote the expression of related genes. APS can enhance the immune function by increasing the weight of immune organs. It has been reported that APS administration (220 mg/kg) in feed could significantly increase the relative weight of immune organs, as well as the number of lactobacilli and *Bacillus cereus* in the intestinal microbiota of chicks [60]. Furthermore, APS could increase the weights of the thymus and spleen, as well as the number of peritoneal macrophages in mice [40], and enhance the function of macrophages. It has been reported that APS at a concentration of 300 µg/mL could significantly increase the nitric oxide (NO), interleukin (IL)-1β, IL-6, and tumor necrosis factor (TNF)-α levels in the human monocyte/macrophage strain TPH-1, indicating that APS can activate macrophages [61]. *Astragalus* polysaccharides showed an impact on the functional status and phenotype of T cells during polymicrobial sepsis. Treatment of mice with APS at a dose of 400 mg/kg on day one after cecal ligation and puncture could increase the T helper (Th) cell population and also the percentage of Th17 cells in the blood. Consequently, APS could attenuate immunosuppression in polymicrobial sepsis [62]. In tilapia fish, it has been shown that 1500 mg/kg APS supplementation could upregulate the phagocytic activity, as well as the superoxide dismutase, glutathione peroxidase, and amylase activities. However, APS had no effect on the serum NO and malondialdehyde levels [63]. In addition, APS can be used as an immunomodulator of vaccines. At a dose of 5, 10, and 20 mg/kg, APS could markedly increase the titer of foot-and-mouth disease virus (FMDV)-specific antibodies in a dose-dependent manner and upregulate the mRNA expression of interferon (IFN)-γ and IL-6, indicating that APS can protect against FMDV [64]. In summary, APS can be used as an immunopotentiator, affecting the non-specific and specific immune systems.

### 6.2. Antitumor Effects

Studies have suggested that the antitumor mechanism of APS may be related to its immune enhancement effect. The polysaccharide could enhance the proliferation of spleen lymphocytes, which explained the stimulation of immune activities observed in rats with stomach cancer [65]. It has been reported that bladder cancer was significantly reduced in mice treated with 50 mg/mL APS, compared with that in the controls, because APS could enhance the innate immune response of bladder epithelial cells by increasing the Toll-like receptor 4 expression [66]. It has also been reported that APS could reduce the telomerase activity and induce the apoptosis of human leukemic HL-60 cells, thus exerting an antitumor effect [67]. Xu et al. [68] have reported that APS showed no direct antitumor effect; however, an antitumor effect was achieved by promoting the production of TNF-α in macrophages and INF-γ in splenocytes. Additionally, APS could inhibit the invasion of HepG2 hepatoma cells by regulating the tumor growth factor-β/Smad signal transduction pathway [69]. In vitro studies do not fully reflect the in vivo antitumor activity of APS. However, the effects of APS on the tumor cell cycle, angiogenesis, telomerase activity, signal transduction, and immune function can all contribute to its antitumor activity.

### 6.3. Anti-Inflammatory Effects

Inflammation is closely associated with immunity, and the inflammatory response mostly involves the immune mechanism. Early symptoms of inflammation, such as increased vascular permeability, inflammatory exudation, and tissue swelling, accompanied by increased levels of inflammatory transmitters, indicate that the immune function of relevant cells has been affected. The *Astragalus* polysaccharides can regulate the signal transduction of nuclear factor-kappa B (NF-κB) and secretion of anti-inflammatory and proinflammatory factors, ultimately balancing the immune response [70]. Zhang et al. [71] have reported that inhibition of adhesion between inflammatory cells and microvascular endothelial cells by downregulating the expression of CD34 on the surface of microvascular endothelial cells may be one of the anti-inflammatory mechanisms of APS. It has been reported that APS could significantly reduce the serum NO level and improve chronic inflammation caused by NO metabolic disorder [72]. At a dose of 200 mg/kg, APS could significantly improve 2,4,6-trinitrobenzene sulfonic acid-induced colitis in rats by downregulating the expression of TNF-α and IL-1β at both mRNA and protein levels, and upregulating the expression of nuclear factor of activated T cells 4 mRNA and protein [73]. Thus, APS can interfere with various inflammatory diseases and affect several pathways and mediators of inflammation. Although research on the anti-inflammatory mechanism of APS has been conducted at the cellular and molecular levels, in-depth studies on its targets are still lacking.

### 6.4. Antiviral Effects

The *Astragalus* polysaccharides can protect the body against viruses, induce, to a certain extent, the production of IFN, and inhibit the viral reproduction. It can also induce CD4$^+$ T cells to produce IL-4, IL-2, and IFN-γ, suggesting that APS can be a potent adjuvant for a hepatitis B virus (HBV) DNA vaccine [74]. It has been reported that 0.5 mg of APS can significantly enhance the efficacy of FMDV vaccine by significantly enhancing the phagocytic capacity of peritoneal macrophages, proliferation of splenic lymphocytes, the titer of serum antibodies, and the production of IL-4 and IL-10. The *Astragalus* polysaccharides can maintain the health of livestock and poultry by inhibiting the propagation of the virus [75]. Moreover, APS could significantly increase the resistance of 15-day-old chickens to H5N1 avian influenza virus [76], and prevent porcine circovirus infection by decreasing the oxidative stress and activating the NF-κB signaling pathway [77]. Furthermore, APS at a concentration of 30 μg/mL could inhibit the reproductive capacity of herpes virus, thereby reducing the incidence of tumor [78]. Briefly, the antiviral activity of APS is generally closely related to cytotoxic T lymphocytes, induced T lymphocytes (CD3$^+$ and CD4$^+$), and NF-κB.

### 6.5. Other Activities

Recent studies on the pharmacological activities of APS have reported that, besides immunomodulatory, antitumor, anti-inflammatory, and antiviral effects, it also exerts antioxidant, anti-aging, cardiovascular protective, antidiabetic, and intestinal protective effects. These pharmacological activities and mechanisms of action of APS are listed in Table 2.

**Table 2.** Pharmacological activities and mechanisms of *Astragalus* Polysaccharides.

| Pharmacological | Experimental Model | Dosage | Mechanism | Reference |
|---|---|---|---|---|
| Antioxidant | Radiation injury model mice | 80 mg/kg | Significantly increased peripheral blood leucocyte count and DNA content in marrow cells, and the activities of SOD in serum. | Liu et al. [79] |
| | Carbon tetrachloride-induced hepatocyte | 200, 400, 800 μg/mL in vitro; 1.5, 3 g/kg in vivo | Inhibited the elevation of GPT, GOT, LDH and MDA; significantly increased the level of SOD | Jia et al. [80] |
| | Human cardiac myocytes | 200 μg/mL | Significantly inhibited generation of ROS | Zhang et al. [81] |
| | Subcutaneously inoculated viable B16-F10 cells male mice | | Significantly inhibited the NBT reduction index | Li et al. [82] |
| | BPD cell model | 2.5 mg/mL | Down-regulated the expression of IL-8, ICAM-1, and NF-κB p65 | Huang et al. [83] |
| Anti-aging | Mouse liver | 100, 200, 300 mg/kg | Scavenging reactive oxygen species (ROS); inhibiting mitochondrial PT; increasing the activities of antioxidases | Li et al. [84] |
| | Aging model of D-galactose mice | 50, 100, 200 mg/kg | Increased the spleen and thymus indexes, and decreased MDA content and increased SOD, GSH-Px, and CAT activity. | Zhong et al. [85] |
| | HDF cell | 1.0 mg/mL | Enhanced cell viability and decreased the number of SAβ-gal positive cells | Zhao et al. [86] |
| Cardiovascular protection | Rat cardiomyocyte injury model; mouse heart failure model | 1.5 g/kg | Restored normal autophagic flux; regulated the AMPK/mTOR pathway | Cao et al. [87] |
| | THP-1 derived foam cells | 25, 50, 100 mg/L | Protected ABCA1 against the lesion of TNF-α in THP-1-derived foam cells | Wu et al. [88] |
| | SD neonatal rat | 10 mg/L | Decreased the expression of ANP mRNA, TNF-α, and IL-6 in extracellular fluid | Zhang et al. [89] |
| | Human cardiac microvascular endothelial cells (HCMEC) | 2.5, 5, 10 mg/mL | Reduced the expression of ICAM-1 and VCAM-1 in HCMEC, inhibiting leukocytes infiltration | Chen et al. [90] |
| Diabetes treatment | KKAy female mice | 700 mg/kg | Restored insulin-induced protein kinase B Ser-473 phosphorylation; translocate glucose transporter 4 in skeletal muscle | Liu et al. [91] |
| | Rat model of type II diabetes mellitus | 700 mg/kg | Restored the glucose homeostasis; reduced the ER stress in the rat model of T2DM | Wang et al. [92] |
| | NOD mice | 2.0 g/kg | Correcting the imbalance between the Th1/Th2 cytokines | Chen et al. [93] |
| Intestinal protection | Ulcerative colitis rat | 200 mg/mL | Increased the volatile fatty acids; and liver bacterial translocation was in effective control; effectively control bacterial translocation in liver | Liang et al. [94] |
| | 2, 4, 6-trinitrobenzene sulfonic acid-induced colitis rat model | 400 mg/kg | Restoring the number of Treg cells; Inhibiting interleukin IL-17 | Zhao et al. [95] |
| | Tumor-bearing mice; γδT cells | 150 and 300 mg/kg | Improved proliferation and function of intestinal intraepithelial γδT cells | Sun et al. [96] |
| Hepatoprotective | Chronic liver injury male SD rats | 450 mg/kg | Lowered the levels of serum ALT, AST, ALP, and hepatic MDA concentration; higher SOD, CAT activities, and GSH concentration | Yan et al. [97] |
| | Liver injury mice | 100 mg/kg | Anti-apoptosis pathway | Liu et al. [98] |
| | CCl4 induced liver damage mice | 1.0 g/kg | Scavenge free radicals to ameliorate oxidative stress and to inhibit lipid peroxidation | Pu et al. [99] |

Note: SOD: Superoxide dismutase; GPT: Glutamate pyruvate transaminase; GOT: Glutamate oxalate transaminase; LDH: Lactate dehydrogenase; MDA: Malondialdehyde; ROS: Reactive oxygen species; NBT: Nitroblue tetrazolium; PT: Permeability transition; GSH-Px: Glutathione peroxidase; CAT: Catalase; AMPK/Mtor: Adenosine monophosphate-activated protein kinase/mammalian target of rapamycin; HCMEC: Human cardiac microvascular endothelial cells; ER: T2DM: ER: Endoplasmic reticulum; T2DM: Type II diabetes mellitus; ALT: Alanine aminotransferase; AST: Aspartate transaminase; ALP: Alkaline phosphatase SD: Sprague-Dawley; BPD: Bronchopulmonary dysplasia; HDF: Human diploid fibroblasts; $CCl_4$: C arbon tetrachloride.

## 7. Conclusions

After several decades of extensive research, great progress has been made in the study of APS. The extraction methods and extraction rate of APS have been continuously improved. It has been found that water, microwave-assisted, ultrasonic wave, enzymatic hydrolysis, and other extraction methods can be combined to improve the extraction rate of APS. Depending on the extraction method and the degree of purification, the chemical composition and structure of APS can be confirmed by HPLC, GC, MS, and NMR. The chemical composition of APS mainly includes glucose, rhamnose, galactose, arabinose, xylose, mannose, glucuronic acid, and galacturonic acid. However, the monosaccharide composition and the structure of sugar chains of APS, obtained by different extraction and purification methods, are different. Therefore, the extraction and purification methods of APS need to be improved continuously.

As an important bioactive component of Astragali Radix, APS shows important pharmacological activities, including immunoregulatory, antitumor, anti-inflammatory, antiviral, and other activities. Currently, studies on the modification of APS mainly employ chemical methods, including sulfation, phosphorylation, selenation, and carboxymethylation, and suggest that structural modification can change the pharmacological activity of APS. However, studies on pharmacological activities of APS usually use crude polysaccharides, which does not allow the establishment of the structure–activity relationship. In addition, the molecular mechanisms of pharmacological activities of APS are still unclear, which also limits its further development and application. Therefore, separation and purification of APS should be improved. Subsequently, the structure–activity relationship of APS should be elucidated at the primary and secondary structure levels. Finally, the pharmacological activities and molecular mechanisms of APS should be studied with homogeneously purified APS.

**Author Contributions:** Conceptualization, J.W., J.J., and M.L.; investigation, J.W., L.S., and X.G.; writing—original draft preparation, J.W., X.G., and J.X.; writing—review and editing, J.W., M.Y., and M.L.

**Funding:** This research was funded by the Science and Technology Innovation Guidance Project, Inner Mongolia (No. KCBJ2018040); National Nonprofit Industry Research (No. 201507002); the Fourth National Traditional Chinese Medicine Resources Survey Project (No. Caishe [2017] 66); the China Agriculture Research System (No. CARS-21); the Science and Technology Planning Project of the Inner Mongolia Autonomous Region, China (No. 201701040); the Standardization Project of Mongolian Medicine in Inner Mongolia (No. 2018-[008]); the Inner Mongolia Natural Science Foundation Project (No. 2018LH03028); the Baotou Science and Technology Project (No. CX-2016-17); and the Scientific Research Foundation Project of Baotou Medical College (No. BYJJ-DF 2017-01).

**Acknowledgments:** The authors are thankful to Na Zhang from Baotou Medical College, who taught us how to use the ChemDraw software.

**Conflicts of Interest:** The authors declare no conflicts of interest.

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
