# Peer review of "Extraction, Structure, and Pharmacological Activities of Astragalus Polysaccharides"

_applsci, doi:10.3390/app9010122_

Round 1

Reviewer 1 Report

This review is very complete and have an interest for the readers. However, biological activities part could be further improve by citing the following papers (see below) and develop a specific paragraph on the hepatoprotective action of Astragalus polysaccharides.

-          Hou et al. 2015 Modulatory Effects of Astragalus Polysaccharides on T-Cell Polarization in Mice with Polymicrobial Sepsis

-          Chen et al 2005 Astragalus polysaccharides: an effective treatment for diabetes prevention in NOD mice.

Author Response

Response to Reviewer 1 Comments

Point 1: Biological activities part could be further improve by citing the following papers (see below)

Hou et al. 2015 Modulatory Effects of Astragalus Polysaccharides on T-Cell Polarization in Mice with Polymicrobial Sepsis

Chen et al 2005 Astragalus polysaccharides: an effective treatment for diabetes prevention in NOD mice.

Response 1: We have added these two references. You can find it in line 283-242, and table 2. (in red)

Point 2: Develop a specific paragraph on the hepatoprotective action of Astragalus polysaccharides.

Response 2: We have added the hepatoprotective action of Astragalus polysaccharides in Table 2. Simultaneously, we added three references [101], [102], and [103]. (in red)

Reviewer 2 Report

The authors have reported some interesting review on the Extraction, structure, and pharmacological activities of Astragalus polysaccharides. The review provides useful information.

Abstract: Abstract is not compatible to the text. May please revise it thoroughly.

The introduction provided extraction procedure, chemical structure and pharmacological activity sounds well.

Author need to give the outline picture of review. Readers easily can understand the review through the outline picture.

conclusion also need to improve.

English of the review needs to be greatly improved. The English of the whole article has to be checked carefully to eliminate linguistic errors.

Author Response

Response to Reviewer 2 Comments

Point 1: Abstract is not compatible to the text. May please revise it thoroughly.

Response 1: We have rewritten the abstract. See the abstract section. (in red)

Point 2: Author need to give the outline picture of review. Readers easily can understand the review through the outline picture.

Response 2: We have made an outline picture of this review. You can find it in the attachment.

Point 3: Conclusion needs to improve.

Response 3: We have rewritten the conclusion. See the conclusion section. (in red)

Point 4: English of the review needs to be greatly improved. The English of the whole article has to be checked carefully to eliminate linguistic errors.

Response 4: We have improved the English for language, grammar, spellings, clarity, and readability of the whole article, and eliminated linguistic errors to the best of our ability.
